# Theory-Guided Design of a Method to Obtain Competitive Balance between U(VI) Adsorption and Swaying Zwitterion-Induced Fouling Resistance on Natural Hemp Fibers

**DOI:** 10.3390/ijms23126517

**Published:** 2022-06-10

**Authors:** Huiquan Gu, Jing Yu, Hongsen Zhang, Gaohui Sun, Rumin Li, Peili Liu, Ying Li, Jun Wang

**Affiliations:** 1Key Laboratory of Superlight Materials and Surface Technology, Ministry of Education, Harbin Engineering University, Harbin 150001, China; 364459961@hrbeu.edu.cn (H.G.); zhanghongsen@hrbeu.edu.cn (H.Z.); lirumin@hrbeu.edu.cn (R.L.); liupeili@hrbeu.edu.cn (P.L.); 2College of Materials Science and Chemical Engineering, Harbin Engineering University, Harbin 150001, China; sungaohui1987@hrbeu.edu.cn; 3Laboratory of Theoretical and Computational Chemistry, College of Chemistry, Jilin University, Changchun 130023, China; liyingedu@jlu.edu.cn

**Keywords:** uranium adsorption, hemp fiber, fouling resistance, zwitterion

## Abstract

The competitive balance between uranium (VI) (U(VI)) adsorption and fouling resistance is of great significance in guaranteeing the full potential of U(VI) adsorbents in seawater, and it is faced with insufficient research. To fill the gap in this field, a molecular dynamics (MD) simulation was employed to explore the influence and to guide the design of mass-produced natural hemp fibers (HFs). Sulfobetaine (SB)- and carboxybetaine (CB)-type zwitterions containing soft side chains were constructed beside amidoxime (AO) groups on HFs (HFAS and HFAC) to form a hydration layer based on the terminal hydrophilic groups. The soft side chains were swayed by waves to form a hydration-layer area with fouling resistance and to simultaneously expel water molecules surrounding the AO groups. HFAS exhibited greater antifouling properties than that of HFAO and HFAC. The U(VI) adsorption capacity of HFAS was almost 10 times higher than that of HFAO, and the max mass rate of U:V was 4.3 after 35 days of immersion in marine water. This paper offers a theory-guided design of a method to the competitive balance between zwitterion-induced fouling resistance and seawater U(VI) adsorption on natural materials.

## 1. Introduction

Uranium (VI) (U(VI)) in seawater is an alternative resource for nuclear energy due to the consequences of the rapidly growing discrepancy between production of and demand for U(VI), as well as the depletion of U(VI) ores projected for in the next century [1,2,3]. Fortunately, representing an unconventional uranium source, the oceans contain about 4.5 billion metric tons of U(VI) [4,5]. To extract U(VI) from seawater, metal–organic frameworks, covalent organic frameworks or biochar, as well as other materials, have been designed and prepared with excellent performance [6,7,8,9,10]. However, the application of the above-mentioned materials is limited by their small size, which makes their use in the sea a challenge [11,12].

Fiber-based adsorbents could overcome the inconvenience of the use of these materials in real seawater [13]. Hemp fibers (HFs) are among the largest produced plant fibers and among the most promising biomaterials for biosorption due to their environmentally friendly qualities and low costs [14,15]. The past decade has seen an intense interest in hemp-based biosorbents [16]. Nevertheless, the application of HFs in seawater is obstructed by the ultralow concentration of U(VI) (~3.3 μg·L^−1^) and various coexistent interfering cations [17,18,19]. The amidoxime (AO) group has been thoroughly studied and utilized to improve the selectivity for seawater U(VI) adsorption in the last half century [20,21,22]. The AO group can coordinate U(VI) ions and form different stable structures (Figure 1a), which could be grafted on HFs to enhance seawater U(VI) adsorption performance [23,24,25,26,27]. However, even with the benefit of the AO group, U(VI) adsorption from seawater is still a long-term process and is significantly limited by the biofouling caused by marine life [28,29].

To eliminate the biofouling influence on seawater U(VI) adsorbents, several fouling-resistance groups have been modified into substrates, such as guanidine, quaternary ammonium, and phosphate zwitterion [30,31,32]. The sulfobetaine (SB) zwitterionic group can immediately form a strong hydration layer around the terminal SO_3_^−^ group in an aqueous environment [33,34]. After having been immersed in marine water, the SB-modified materials are covered by the instantly formed hydration layer ahead of the formation of the organic conditioning film, which is the foundation of the following three steps of marine fouling [35,36,37]. In addition, the hydration layer is also considered as a biofriendly strategy for fouling resistance by antiadhesion, instead of cell-membrane damage [38,39,40]. The fouling resistance of SB has been studied by molecular dynamics (MD) simulation, as well as carboxybetaine (CB), to instruct the design of novel materials with antifouling properties [41]. However, to the best of our knowledge, the influence of a zwitterion-induced hydration layer on U(VI) adsorption has not been investigated so far.

In this study, several models were built as candidates for MD simulation to investigate the effect of the hydration layer on U(VI) adsorption. The simulated results were converted into real adsorbents by the construction of SB or CB zwitterionic groups onto swaying soft side chains around AO groups on HFs, namely HFAC and HFAS, respectively. Density functional theory (DFT) calculation was utilized to understand the coordinated form of U(VI) ions and AO groups on HFs. The adsorption performances of HFAO, HFAC, and HFAS were investigated by batch U(VI) adsorption experiments in the laboratory and 35 days of immersion in marine water (Yellow Sea, Dalian, China). This paper presents the theory-guided design of a novel method to obtain the competitive balance between zwitterion-induced fouling resistance and seawater U(VI) adsorption on mass-produced natural fibers.

## 2. Results and Discussion

### 2.1. MD Simulation of AOSB0, AOSB1, and AOSB2

The above-mentioned models were simulated by MD calculation, and the structural formulas are given in Figure 1c and Appendix A. Five cellulose molecules with only one AO group at the center were named AOSB0; AOSB1 and AOSB2 were built based on AOSB0, with one and two SB-type zwitterions, respectively, grafted onto HDI to construct soft side chains. AOCB1 and AOCB2 were also built similarly to AOSB1 and AOSB2, except for the terminal COO^−^ group. The radial distribution functions (RDFs) of water molecules around the AO group in AOSB0, AOSB1, and AOSB2 are shown in Figure 1a, and those for AOCB1 and AOCB2 are shown in Appendix A, presenting the same peak positions at 0.28 nm. This result indicated that the introduced swaying soft side chains showed no impact on the position of water molecules around the AO groups. The dispositions of water molecules around the AO groups in AOSB0, AOSB1, and AOSB2 are exhibited in Figure 1c, and those for AOCB1 and AOCB2 are exhibited in Appendix A. There were 14 water molecules surrounding the AO group in AOSB0 in the absence of swaying soft side chains. After having been modified by soft side chains, the number of water molecules around the AO group was reduced with the increase in the amount of side chains, which was 10 and 6 for AOSB1 and AOSB2, respectively, and 14 and 6 for AOCB1 and AOCB2, respectively. The blue clouds in the models stand for the hydration layer around the terminal SO_3_^−^ or COO^−^ group. Thus, the introduced swaying soft side chains were able to reduce the number of water molecules around the AO groups. The interaction energy of each model in Figure 1c shows that AOSB1, at 5.8 ns, interacted with U(VI) ions for half the time needed by AOSB0, at 11.5 ns, in a whole period of 50 ns. Therefore, fewer water molecules around the adsorption site accelerated the adsorption kinetics by reducing the competition between water molecules and U(VI) ions for adsorption sites. The electrostatic interaction energy and the van der Waals (vdW) interactions between U(VI) ions and the AO groups (from both AOSB0 and AOSB1) were stable at about −600 and 150 kJ·mol^−1^, respectively. The negative values of interaction energy indicated the stronger attraction of the U(VI) ions [42]. However, the AO group on AOSB2 had almost no interactions with U(VI) ions under the same conditions. Compared with AOSB1, the two side chains on AOSB2 exhibited larger hydration-layer areas during the swaying process, which limited the contact between U(VI) ions and the AO group. In addition, the steric hindrance from the two swaying soft side chains should also be considered. AOCB2 adsorbed U(VI) ions after a longer period, which might be due to the weaker hydration layer formed by the terminal COO^−^ group. Thus, the appropriate amount of swaying soft side chains containing SB-type zwitterions is a benefit for U(VI) adsorption onto HF-based materials, according to the theoretical results from the MD simulation.

### 2.2. Characterization of HFAO, HFAC, and HFAS

The SEM images of HFs, HFAO, HFAC, and HFAS in Figure 2a show that all the HF-based materials remained fibrous in structure. The sample of HFs showed the smoothest surface among all the samples. Some etches on the surfaces of HFAS and HFAC could be observed after further modification. Fortunately, the backbones of HFAO, HFAC, and HFAS were not damaged by the several steps of chemical modification. The Fourier transform infrared (FTIR) spectroscopy and X-ray photoelectron spectroscopy (XPS) spectra of the HF-based adsorbents are listed in Figure 2c,d. Compared with the spectra of the HFs, the stretching vibration of C≡N on HFCN showed at 2251 cm^−1^, indicating the successful modification by acrylonitrile [43]. The peak of C≡N groups disappeared in the spectra of HFAO, HFAC, and HFAS after amidoximation (Figure 2c). The new peaks at 1772 cm^−1^ (HFAC) and 1787 cm^−1^ (HFAS) were due to C=N-O groups, indicating the successful introduction of HDI onto the hemp fibers. From the spectra of HFAS, the peak at 1202 cm^−1^ suggested the modification of SO_3_^−^ groups [44]. All the HF-based adsorbents were further analyzed by XPS tests to determine the surface chemical bonding status (Figure 2d). The higher N element content of HFAC and HFAS compared with HFAO revealed by XPS spectra indicated that HDI and DMEA were successfully modified onto HFAC and HFAS. Furthermore, the XPS survey scan spectra of HFAS proved the existence of S 2p (168.1 eV) and S 2s (231.1 eV), which was indicative of atoms from sulfobetaine [45,46].

### 2.3. Batch U(VI) Adsorption Experiments with HFAO, HFAC, and HFAS

The U(VI) adsorption kinetics of HFAO, HFAC, and HFAS were investigated. The equilibrium times of both HFAS and HFAC were 2 h, which was 1 h faster than that of HFAO at pH = 8.3 (Figure 3). This result was consistent with the theoretical results obtained from the MD simulations. The accelerated adsorption kinetics of HFAC and HFAS were also obtained at optimized pH and are shown in Appendix A. In order to explore the essential mechanism from a macro perspective, HFAO, HFAC, and HFAS were further tested by water contact angle (WCA) and total permeant time (TPT) (Figure 3e and Appendix A). All the HF-based adsorbents exhibited fast permeated processes after the touch of water drops on the surface, due to the abundant inherent -OH groups on the HFs and further modified groups. The WCAs of HFAO showed to be slightly greater than those of the HFs, and the TPTs were also 0.2 s longer. The introduction of CB- and SB-type zwitterions improved the hydrophilicity of the substrate and shortened the TPTs of HFAC and HFAS by about half (1.4 s) and a quarter (0.8 s), respectively. The accelerated contact between the adsorbents and water molecules benefitted the diffusion of U(VI) ions onto the active sites of the absorbents [47]. Combining the lessened water molecules around the adsorption sites, the adsorption equilibrium time was shortened from 3 h (HFAO) to 2 h (HFAC and HFAS). The low WCAs and fleeting TPTs of HFAC and HFAS would be an advantage for applications in marine environments, because the first biofouling film would be formed within several seconds after immersion. Following the above results, the dynamic processes were further analyzed by pseudo-1st-order, pseudo-2nd-order, and Weber–Morris (W-M) kinetic equations [48]. From Figure 3 and Appendix A and Appendix A, it can be seen that the adsorption kinetics of the HF-based adsorbents were closer to the pseudo-2nd-order model with R^2^ > 0.99. In addition, the values of Q_e,cal_ gained from the pseudo-2nd-order model were much closer to the experimental results. The above results suggest that the rate-determining step of all the HFAO, HFAC, and HFAS was mainly chemical extraction. From Appendix A, it can be seen that the values of C following the three steps of HFAO, HFAC, and HFAS indicated that the edges of HFAC and HFAS made a larger contribution to adsorbing U(VI) than HFAO at both optimized pH and pH 8.3. In addition, none of the W-M linear fitted curves passed through the origin, meaning there was no unique rate-limiting step. HFAO, HFAC, and HFAS showed a three-step U(VI) extraction behavior. U(VI) ions went across the aqueous film to the surface of the absorbent, entered the interior of the materials, and chelated the organic functional groups. In Appendix A and Appendix A, we show the assessment of the U(VI) adsorption isotherm of HFAO, HFAC, and HFAS at both pH 8.3 and optimized pH at 25 °C, fitted according to the Langmuir, Freundlich, and Dubinin–Radushkevich (D-R) models. The results show that the isotherms fitted closer to the Langmuir model (R^2^ > 0.99) than to the Freundlich and D-R models at both pH 8.3 and optimized pH at 25 °C. The values of the experimental results for HFAO, HFAC, and HFAS were close to the Q_m,cal_ of the Langmuir model (Appendix A). Corresponding with the kinetics results, the U(VI) extracting behavior of HFAO, HFAC, and HFAS was that of monolayer chemisorption. In addition, the binding energy on the entire surfaces of HFAO, HFAC, and HFAS was uniformly and homogeneously distributed.

The stability and reusability of the adsorbent is a significant factor to assess the practical application value of adsorbents. Hence, five-time adsorption–desorption cycles were employed to prove the regeneration ability of HF-based materials. The eluting efficiencies of HNO_3_, citric acid, EDTANa_2_, NaHCO_3_, and NaOH at 0.1 mol·L^−1^ are compared in Appendix A. Among them, HNO_3_ was selected as the eluent and utilized in the adsorption–desorption cycle experiments due to the largest desorption percentage (86.6 ± 3.2%). In Appendix A, the adsorption rate of the HF-based materials is seen to decrease due to the adsorption–desorption procedure, but the adsorption efficiencies of HFAS and HFAC still remained 83.0 ± 4.2% and 82.4 ± 2.2% after five cycles. In summary, the stability and reusability of HFAS proved their promising value as seawater U(VI) materials.

### 2.4. The Adsorption Performance of HF-Based Materials in Ion Competition Solution, U(VI)-Spiked Nitzschia closterium Solution, and Seawater

After the batch experiments at high concentrations, HFAO, HFAC, and HFAS were added into simulated seawater with concentrations of U(VI), V(V), Fe(III), Co(II), Ni(II), Cu(II), Zn(II), and Pb(II) 100 times those in seawater (Figure 4a and Appendix A). In addition, U(VI)-spiked *Nitzschia closterium* (*N*. *closterium*), *Phaeodactylum tricornutum* (*P*. *tricornutum*), and *Halamphora* sp. solutions at ~1 mg·L^−1^ were prepared to simulate the adsorption performance under the influence of marine life (Appendix A). The antifouling properties of HFAO, HFAC, and HFAS were investigated by having them immersed in *N*. *Closterium*, *P*. *tricornutum*, and *Halamphora* sp. solutions for 2 and 7 days under both 12 h:12 h light:dark cycles and dark conditions to simulate the adhesion of marine organisms (Appendix A). G^+^ strain *Staphylococcus aureus (**S. aureus**)*, G^−^ strain *Escherichia coli* (*E. coli*), and marine bacteria were also selected to test the antifouling properties of HFAO, HFAC, and HFAS, as shown in Appendix A. Finally, HFAO, HFAC, and HFAS were subjected to 35 days of immersion in the Yellow Sea, China (Figure 4b). In the ion competition experiments, HFAS exhibited the highest U(VI) adsorption capacity of 624.3 ± 48.2 μg·g^−1^, but the lowest V(V) adsorption capacity of 242.3 ± 14.3 μg·g^−1^ among the HF- based materials, leading to the max mass rate of U:V at 3.0. The distribution coefficient ($K_{d}^{Ion}$) values of each of the ions were calculated and are given in Appendix A to demonstrate the affinity of HFAO, HFAC, and HFAS. The $K_{d}^{U}$ of HFAS was 8.2 ± 2.6 L·g^−1^, illustrating the great U(VI) affinity in the presence of other interfering ions. Appendix A shows the influence of marine life on the adsorption performance of all three HF-based materials simulated by the three diatoms. HFAC and HFAS exhibited higher adsorption capacities than HFAO, due to the introduction of zwitterionic groups. The U(VI) adsorption capacities of HFAS were 1819.2 ± 49.7, 2024.9 ± 53.1, and 1887.3 ± 58.8 μg·g^−1^, which were almost twice those of HFAO (1000.3 ± 43.8, 932.3 ± 89.0, and 828.2 ± 64.4 μg·g^−1^, respectively) in the U(VI)-spiked *N*. *Closterium*, *P*. *tricornutum*, and *Halamphora sp.* solutions after 48 h. The U(VI) adsorption capacities of HFAO, HFAC, and HFAS were 1801.2 ± 25.2, 1943.8 ± 52.9, and 2064.3 ± 43.3 μg·g^−1^, respectively, in the U(VI)-spiked simulated seawater after 48 h. The smaller drop in the adsorption capacities of HFAC and HFAS with respect to HFAO was due to the fact that the zwitterion side chains endowed the materials with antifouling properties to reduce the influence of diatoms. This result also indicated that the antifouling properties of HFAS were greater than those of HFAC. The influence of these diatoms was tested by immersing the HF-based materials in the three kinds of diatoms solutions. The fluorescence microscopy images of the HFs, HFAO, HFAC, and HFAS immersed for 2 days, displayed in Appendix A, show that diatom cells only adhered to the HFs and HFAO under both light–dark cycle and dark conditions. After soaking for 7 days, more diatoms cells could be observed on the HFs and HFAO under both conditions, indicating that the original HFs and the modified AO groups had no antifouling properties. According to the above results, the adhesion of the three kinds of diatom cells hindered the interaction between HFAO and the U(VI) ions, leading to reduced adsorption capacity. For HFAS, no cells adhered on the surface even after 7 days of immersion in the three kinds of diatom solutions. Furthermore, the great antifouling properties of HFAS were also proved by antibacteria tests, shown in Appendix A. The above results point at the long-lasting and broad-spectrum antifouling properties conferred to HFAS by the surrounding hydration layer. Finally, HFAS exhibited the most affinity and the highest adsorption capacity for U(VI) (184.8 ± 18.4 μg·g^−1^) among the three HF-based adsorbents, which was almost 10 times that of HFAO (18.3 ± 2.5 μg·g^−1^) after immersion in the Yellow Sea, China, for 35 days. HFAS also showed the max mass rate of U:V of 4.3, which was higher than that obtained in the laboratory, due to the inexhaustible U(VI) ions in marine water. The higher adsorption capacity of HFAS with respect to HFAC observed during the marine tests was due to the greater antifouling properties. However, the lower adsorption capacity of HFAS in marine water was because of the complex marine environment, such as numerous marine organisms and different concentrations of coexisting ions. These experimental results further demonstrated that antifouling properties are vital for the application of adsorbents in an ocean environment and prove that HFAS have greater potential than HFAC as seawater U(VI) adsorbents.

### 2.5. Analysis of U(VI) Adsorption on HFAS

The coordinated form of U(VI) ions and HFAS was studied by XPS and density functional theory (DFT) calculation. The XPS survey spectrum of HFAS after U(VI) adsorption (HFASU) and the U 4f high-resolution spectrum are given in Figure 5a,b, respectively. Two peaks at 392.5 and 381.7 eV with 10.8 eV splitting were observed and were attributed to the typical peaks of U 4f_5/2_ and U 4f_7/2_, respectively [49]. N 1s and O 1s high-resolution spectra, analyzed to investigate the coordinated form of HFASU, are displayed in Figure 5c,d, respectively. The N 1s high-resolution spectrum could be curve-fitted with four peaks at 399.3, 400.3, 401.2, and 402.5 eV for H_2_N-C=**N**-OH, C-**N**H_2_, **N**-COO, and **N**^+^(CH_3_)_2_${CH}_{2}^{-}$, respectively [44,46,50]. After the extraction of U(VI) onto HFAS, the peaks of H_2_**N**-C=N-OH and C-**N**H_2_ reached 399.4 and 400.4 eV, respectively. The O 1s high-resolution spectra in Figure 5d shows O=C-**O**, C-**O-**C/C-**O**H, H_2_N-C=N-**O**H, and **O**=C-O peak at 534.0, 533.1, 532.2, and 531.1 eV, respectively. The H_2_N-C=N-**O**H peak changed to 532.3 eV after the adsorption of U(VI). After the adsorption of U(VI), all the binding energy shifted to higher values, indicating the decrease in electron density around the atoms [51,52].

DFT calculation was employed to understand the contribution of N and O atoms from the AO group in depth. The resulting six structures showed lower binding energy (*E*_ads_) than UO_2_(CO_3_)_3_^4−^ (−31.8 eV), as shown in Figure 6 and Appendix A, indicating greater stability than UO_2_(CO_3_)_3_^4−^ and the trend of coordination between the adsorption sites and U(VI) ions [53,54]. AO(UO_2_)(CO_3_)(H_2_O) exhibited the lowest *E*_ads_ (−38.8 eV) among the six structures, and it chelated with the U(VI) ion by the N atom from the amino group and the O atom from the oxime part in the AO group. Other structures coordinated U(VI) ions only with the oxime parts. *η*^2^-AO(UO_2_)(CO_3_)(H_2_O)_2_-1, *η*^2^-AO(UO_2_)(CO_3_)(H_2_O)_2_-2, *η*^2^-AO(UO_2_)(CO_3_)(H_2_O), and *η*^2^-AO(UO_2_)(CO_3_)_2_ replaced two CO_3_^2−^ from UO_2_(CO_3_)_3_^4−^ to coordinate U(VI) ions, and water molecules from the aqueous solution stabilized the coordination structures by direct participation or hydrogen bonds. In the structure of *η*^1^-AO(UO_2_)(CO_3_)_2_, only one O atom from the oxime part bound with a U(VI) ion. The higher *E*_ads_ of *η*^2^-AO(UO_2_)(CO_3_)_2_ and *η*^1^-AO(UO_2_)(CO_3_)_2_ with respect to the other structures may be because two CO_3_^2−^ in the coordination structure occupied the equatorial ring of the U(VI) ion and did not leave much space for the AO group. In addition, the AO group on cellulose may have led to steric hindrance in the structures. Thus, both the XPS analysis and DFT calculation results proved that the AO groups on HFAS were the main U(VI) adsorption groups. In this case, the high affinity of HFAS for U(VI) was due to the introduction of the SB-type zwitterionic group containing soft sides, which improved the hydrophilicity of the material but locally expelled the water molecules surrounding the AO group, which, in turn, reduced the competition between water molecules and the U(VI) ions and increased the opportunity of contact between the H_2_N-C=**N**-**O**H groups and U(VI) ions.

## 3. Materials and Methods

### 3.1. Amidoxime-Modified Hemp Fibers (HFAO)

Hemp fibers were pretreated as previously described [55]. In total, 9.0 g of treated HFs, 1.8 g of tetrabutylammonium bromide (TBAB), 216 mL of acrylonitrile (AN), 6 mL of deionized (DI) water, and 6 mL of 36% NaOH solution were added into a round-bottom flask, stirred at 25 °C for 0.5, 1.0, 1.5, 2.0, and 2.5 h, and named HFCN_0.5–2.5_. After the reaction, the solution was filtered. DI water and ethanol were used to wash all the HFCN_0.5–2.5_, which were dried overnight at 60 °C in an oven. In total, 6.0 g of HFCN_0.5–2.5_, 20.0 g of hydroxylamine hydrochloride (NH_2_OH·HCl), 15.2 g of Na_2_CO_3_, and 200 mL of DI water were placed in a round-bottom flask and stirred at 70 °C for 6 h. DI water and ethanol were used to wash all the resulting HFAO_0.5–2.5_, which were also dried overnight at 60 °C in an oven. The adsorption capacities of HFAO_0.5–2.5_ at different pH values (4.0–9.0) were estimated (Appendix A). From Appendix A, it can be seen that the adsorption capacities of HFAO_2.0_ and HFAO_2.5_ were similar at pH = 8.3. Thus, 2.0 h was selected to prepare HFAO.

### 3.2. Zwitterion-Modified Hemp Fibers (HFAC and HFAS)

In total, 0.5 g of HFAO, 0.46 g of hexamethylene diisocyanate (HDI), a catalytic dosing of dibutyltin dilaurate (DBTDL), and 40 mL of anhydrous N,N-dimethylformamide (DMF) were placed in a round-bottom flask and stirred at 50 °C for 0.5, 1.0, 1.5, 2.0, and 2.5 h under nitrogen. After completion, the reaction was washed with anhydrous DMF. The obtained fibers were placed in a round-bottom flask with 40 mL of anhydrous DMF, 0.62 g of N,N-dimethylethanolamine (DMEA), and a catalytic dosing of DBTDL. Then, the solution was stirred at 50 °C for 2.5 h in an N_2_ atmosphere. DMF, DI water, and ethanol were used to wash all the resulting fibers, followed by drying them overnight at 60 °C in an oven. A total of 0.5 g of the above obtained fiber, 0.66 g of 1,3-propanesultone (PrS) or 0.68 g of 1,4-butyrolactone (1,4-BL), and 20 mL of ethanol were placed in a round-bottom flask and reacted at 60 °C for 2 h [56]. DI water and ethanol were used to wash all the resulting fibers, followed by drying them overnight at 60 °C in an oven. Zwitterionic-groups-modified hemp fibers, HFAC_0.5–2.5_ and HFAS_0.5–2.5_, were obtained and tested under the same condition as HFAO. In Appendix A, it is shown that 0.5 h was necessary to prepare HFAC and HFAS, due to the appropriate number of zwitterion soft side chains around the AO groups on the HFs. The synthesis routes of HFAO, HFAC, and HFAS are schematically illustrated in Figure 1b.

## 4. Conclusions

In this study, AO groups were grafted onto HFs (HFAO) to enhance the selectivity for U(VI), and sulfobetaine (SB) and carboxybetaine (CB) zwitterion side chains were further grafted on HFAO to endow the material with antifouling properties. The SB and CB side chains formed the hydration layer that protected the materials from fouling organisms and expelled water molecules around the AO groups to improve their hydrophilicity and adsorption properties. The results of the antifouling tests from *N. closterium*, *P.*
*tricornutum*, *Halamphora* sp., *S. aureus*, *E. coli*, and marine bacteria proved that HFAS had better antifouling properties than HFAC and HFAO. The uranium adsorption capacities of HFAS in uranium-spiked simulated seawater, i.e., the *N*. *closterium*, *P*. *tricornutum*, and *Halamphora* sp. solutions, were 2064.3 ± 43.3, 1819.2 ± 49.7, 2024.9 ± 53.1, and 1887.3 ± 58.8 μg·g^−1^, which were 1.1, 1.8, 2.2, and 2.3 times those of HFAO, respectively. The adsorption capacities of balance-optimized HFAS exhibited almost 10 times higher effectiveness than those of HFAO, and the max mass rate of U:V was 4.3 after 35 days of immersion in marine water in China. From the results of the XPS and DFT calculation, the AO groups could coordinate U(VI) ions to form different stable coordination structures. The above results provide new perspectives for the theory-guided design of a method to obtain the competitive balance between zwitterionic-group-induced fouling resistance and U(VI) adsorption capacity in seawater, as well as a basis for the mass production of U(VI) adsorbents based on natural materials.

## Data Availability

Not applicable.

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
