# Peer review of "Theory-Guided Design of a Method to Obtain Competitive Balance between U(VI) Adsorption and Swaying Zwitterion-Induced Fouling Resistance on Natural Hemp Fibers"

_ijms, 2022, doi:10.3390/ijms23126517_

Round 1
Reviewer 1 Report
In this contribution the authors focus on the fabrication of new U(VI) adsorbents from seawater. It is interesting approach, and I liked the idea. However, for accepting this work, manuscript must be significantly improved to fulfil the criteria of IJMS. Due to this, I will highlight only the most problematic parts.
In general manuscript is written in cryptic form and must be cleaned. Reader is confused as logical red line leading reader throught the manuscript is missing. Furthermore, I will appreciate to see more schemes, figures, tables, etc. The critical parts (why I cannot recommend accepting this manuscript in presented form) are highlighted in bold.
- Graphical abstract is missing. It will make easier for reader to understand the aim of the work.
- In the introduction, the structure of the amidoxime-U(VI) complex is described in detail. Please, add the structure as a figure, for clarity.
- I found few incorrect statements and references. For example: page 2 lines 51-53. To eliminate the biofouling influence on seawater U(VI) adsorbents, several fouling resistance groups had been modified to substrates, such as guanidine, quaternary ammonium, and zwitterion. [27-29] It is not fully correct. Only one specific type of zwitterion – phosphorylcholine is referred in reference No. 29. No sulphobetaines or carboxybetaines etc. Zwitterion is not a synonym for the phosphorylcholine.
- Nota bene. Why phosphorylcholine modification of the HF was not challenged in your work? It will be nice to compare your results with ref 29.
- Figure 1. Table with the structures of AOSB0, AOSB1 and AOSB2 is needed as it is not clear from the images. Why the order of images is reversed AOSB0, AOSB2 and only after AOSB1? What do the numbers (14,6,10) bellow the names?
- Please be more careful with reference to the further text. For example, page 3 line 111: … HFAS and HFAC were prepared (details in Section 2). No, there is no „Section 2“. There is section „Materials and Methods“.
- I miss description of chemistry. In the main text, the chemistry is only mentioned. I find phrases like: …after several steps… (page 3, line 113) colloquial. Also, I miss reaction conditions in the figure 2a. Some parts, for example the adsorption experiments or characterisation are nicely discussed, but not the synthesis...
- Figure 2a. Why is modification with carboxybetaine omitted here?
- Paragraph 2.3 is quite complex. Please simplify it. I will recommend focussing exclusively on the results obtained at pH=8.3 in the main text. Rest can be discussed described (additional table, adsorption curves?) in the SI. When you start to describe the adsorption capacity at different pH etc. the red line (and main goal) of the manuscript is lost.
- Figure 4 must be changed; it is not clear what is test solution and what is seawater (the colour separation is barely visible). I miss the statistic! Results are presented without any performed statistical analysis! What are the bars? Are they SEM or SD? It is not even mentioned in the main text. Use of different scales for Qe (y axes) is confusing. In this form, Figure 4, can be considered as hype as it suggests that adsorbents are even more effective in seawater than in the test solution.
- Please do not use decimal numbers in the text (when it is not necessary). It makes it hard to read e.g., page 6 line 239: 1801.20±25.24. If you just write1801±25 it makes no difference and it is easier to read.
- English must be corrected and improved. For example:
- Page 1 lines 34-36: Several kinds of materials with excellent performance have been developed, such as metal organic framework, covalent organic framework or biochar, etc. For what? The objective is missing…
- Page 1 lines 38-41: Fiber-based adsorbents could overcome the inconvenience of disposition in real seawater, such as hemp fiber (HF), which is one of the largest produced plant fibers and one of the most promising bio-materials for biosorption due to its environmentally friendly qualities and low cost. In this sentence two unrelated topics are mixed. It will be much easier to understand, if the sentence is split to: Fiber-based adsorbents (e.g., hemp based adsorbents), could overcome the inconvenience of disposition in real seawater. Hemp fiber (HF) is one of the largest produced plants fibers and one of the most promising bio-materials for biosorption due to its environmentally friendly qualities and low cost.
- Page 2 lines 91-92: Therefore, the fewer water molecules around the adsorption site, the faster adsorption kinetics. The verb is missing.
- Page 9 line 309: Hemp fibers were pre-treated by our previous report. No, fibers were not treated by the report. Correct way will be: Hemp fibers were pre-treated as described previously.
All in all, I will suggest to rewrite the manuscript and start the new submission procedure.
Reviewer 2 Report
In the manuscript entitled “Theory-guided design of competitive balance between U(VI) adsorption and swaying zwitterion induced fouling resistance on natural hemp fibers”, H. Gu et al. have proposed three models for molecular dynamic simulation to investigate the effect of hydration layer on U(VI) adsorption on natural hemp fibers.
First of all, the abstract is too long, I suggest to reduce, indicating briefly the motivation, the aim and the obtained results.
In the introduction, the authors should indicate which are the common methods reported in literature for U(VI) adsorption in seawater, reporting their advantages and disadvantages in comparison with the porposed one. Moreover, the authors could compare and estimate the economic impact of reported procedure and the proposed one.
In a manuscript common structure, the materials and methods section should be reported before the results; please, correct.
In the material and methods section the authors should add a section regarding all the instruments used to characterize the reporting system, indicating also the experimental conditions. Please, add.
In the manuscript the synthetic route used to functionalize the natural fibre should be added indicating all the experimental conditions.
The authors should add more details on MD simulation procedure; they used a program, etc.? Please, indicate.
The English style is quite acceptable.
I canb accept with minor revisions.
Reviewer 3 Report
The english should be slightly improved as many phrases are difficult to follow “ for example “. Thus, the introduced swaying soft side chain showed no impact on the position but the amount of water molecules around AO groups.” other many formulation were found the same difficult
Rather than saying “y immersion in Yellow Sea, China.” It will be better providing actually the water characteristics
Actually from the abstract and introduction is not very clear what the authors propose in this work; right now seems very vague and general approach taken
Why MD simulation and not other tool was proposed
Nothing about boundary conditions
“were prepared (details in Section 2)” very vague actual this line make reference to section 2 but it is included in
You have to make a clear difference between methods and results cause actually it is very difficult to follow this work
Line 285-287 is not clear information provided
The discussion of results are almost none – please elaborate in
The conclusion are very vague and not related to the results
Some recent references are required
Round 2
Reviewer 1 Report
Dear Authors,
I appreciate the improvement of the manuscript, chards, graphical abstract, etc. are now much clearer. Unfortunately, I still cannot recommend accepting this manuscript for publication in current form. Red line is still missing – namely, story about the carboxybetaine (CB) derivative is misleading.
CB is not mentioned in abstract and conclusions. There is missing discussion about it. It appears in the intro and M&M section, but MD simulations are provided only for SB. Why? It is not clear why you completely focus on SB? I assume that the reason is better adsorption capacity of HFAS vs HFAC in sea-water. But this must be clearly discussed.
Also, the I miss discussion why you observe dramatical decrease of adsorption capacity in sea-water vs test solution, e.g. HFAC drops 12x (!) and HFAS 3x. I miss discussion/mental analysis of the data.
Please add discussion/conclusion where you compare CB vs SB vs PS (from your previous publication). What is the final recommendation? What have we learned from all your experiments?
Minor point:
FTIR spectrum for HFCN is provided but we do not see the chemical structure. Instead of repeating the same scheme (scheme 1 and figure 2) you can add the organic chemistry scheme with the reagents & conditions and with the abbreviations bellow the chemical structures. Please use the style for organic chemistry publications. I will recommend using clear font (e.g. Arial) for chemical structures (e.g. ACS or Synlett) and also for chards. It will be much easier to read, and it will look more professional.
Reviewer 3 Report
,
Round 3
Reviewer 1 Report
Dear Authors,
Thank you very much for all your answers and work on the manuscript. Now I can clearly accept it for the publication. The improvement is highly notable and current final version is much better than original one. I think, we both learned a lot during this reviewing process.
Cheers!
VS